# Dual Reasoning: A GNN-LLM Collaborative Framework for Knowledge Graph Question Answering

Guangyi Liu[1], Yongqi Zhang[2], Yong Li[1], Quanming Yao[1*]
[1]Tsinghua University, [2]HKUST (GZ)
`liugy24@mails.tsinghua.edu.cn, yongqizhang@hkust-gz.edu.cn,`
`liyong07@tsinghua.edu.cn, qyaoaa@tsinghua.edu.cn`

Large Language Models (LLMs) excel at intuitive, implicit reasoning. Guiding LLMs to construct thought chains can enhance their deliberate reasoning abilities, but also faces challenges such as hallucination. Knowledge Graphs (KGs) can provide explicit structured knowledge for LLMs to alleviate these issues. However, existing KG-enhanced methods often overlook explicit graph learning, making it challenging to efficiently provide precise reasoning chains for LLMs. Following dual-process theory, we propose Dual-Reasoning (DualR), a novel framework that integrates an external system based on Graph Neural Network (GNN) for explicit reasoning on KGs, complementing the implicit reasoning of LLMs through externalized reasoning chains. DualR designs an LLM-empowered GNN module for explicit learning on KGs, efficiently extracting high-quality reasoning chains. These reasoning chains are then refined to a knowledge-enhanced multiple-choice prompt, guiding a frozen LLM to reason thoughtfully for final answer determination. Extensive experiments on three benchmark KGQA datasets demonstrate that DualR achieves state-of-the-art performance while maintaining high efficiency and interpretability. Our code and data are available in `https://github.com/leolouis14/DualR`.

## 1. Introduction

Large language models (LLMs) [1–4] have demonstrated impressive capabilities across various natural language processing tasks. Pre-trained on extensive corpora, LLMs excel at implicit and associative reasoning. To further enhance LLMs' reasoning abilities, many approaches (e.g., Chain-of-Thought (CoT)[5], Tree-of-Thought (ToT)[6]) guide LLMs to generate intermediate steps and form a complete thought chain, aiming for a more deliberate and explicit reasoning [7, 8]. While these approaches can improve performance, they typically increase inference costs, and encounter the challenge of hallucinations when the model lacks relevant knowledge, especially domain-specific and up-to-date knowledge [7, 9].

Integrating external knowledge sources, such as knowledge graphs (KGs), offers a promising solution to these limitations. KGs, storing a vast amount of facts in the form of triples (e.g., Wikidata [10], YAGO [11], and NELL [12]), are vital for a variety of applications due to their capacity to deliver explicit knowledge [13, 14]. A common and essential task for integrating LLMs with KGs is Question Answering over Knowledge Graph (KGQA), which aims at answering natural language question from entities within a given KG. To accurately respond to a given question, a key challenge is that how to enable LLMs to effectively acquire supportive reasoning evidence from a large and complex KG structure. Existing methods frequently overlook the importance of explicit learning within graph structures, which is essential for supplying precise evidence chain for LLM's reasoning. The text-based retrieval methods (e.g.,KAPING [15]), directly retrieve triplets based on text similarity from KGs for LLMs, which frequently result in redundant or irrelevant information [16]. Another LLM-based retrieval methods like StructGPT [17], ToG [18], guide LLMs to retrieve over KGs across multiple steps. Since LLMs lack the inherent capacity to comprehend graph structures, it is often challenging for them to perform effective topological reasoning on graphs [9]. Furthermore, frequent

---
*Correspondence is to Q.Yao.

Second Conference on Parsimony and Learning (CPAL 2025).

interaction with LLMs entails significant time and resource costs, especially in the KGQA task that requires multi-hop reasoning on a large KG.

To effectively integrate LLMs with KGs and address challenges above, we draw on cognitive science for inspiration. Bengio et al. [19] has highlighted connections between deep learning and the cognitive process described in Daniel Kahneman's *Thinking fast and Slow* [20]. The dual-process theory in the book posits that human thinking arises from two complementary systems: the intuitive and implicit System1, and the deliberate and explicit System 2 [21, 22]. Building on this, we introduce Dual-Reasoning (DualR), a novel approach that incorporates an external, structured "System 2" to explicitly reason on KGs, extracting valuable reasoning chains to complement the intuitive and implicit "System 1"-like reasoning of LLMs. To efficiently implement this externalized "System 2", we utilize graph neural networks (GNNs) [23–25], which are well-suited for learning within complex graph structures. Specifically, we first design an LLM-empowered GNN module to explicitly reason on the KG, extracting high-quality reasoning chains relevant to the question. Then, the identified reasoning chains are refined into a knowledge-enhanced multiple-choice prompt, guiding a frozen LLM to reason thoughtfully for final answer determination. With the collaboration of "System 1" and "System 2", DualR enables efficient and explicit reasoning over KGs, enhancing the reasoning capabilities of LLMs through extracted reasoning chains, and achieving accurate and faithful results in KGQA task. Furthermore, the framework is designed to integrate seamlessly with any off-the-shelf LLMs with just one-step inference, without requiring extensive interactions or fine-tuning, thus allowing for resource-efficient deployment. The contributions are summarized as follows:

- Following dual-process theory, we introduce a novel framework, Dual-Reasoning (DualR), which integrates an external, structured "System 2" for deliberate, explicit reasoning on KGs, complementing the implicit reasoning of LLMs through externalized reasoning chains.

- To implement this framework, we design a lightweight GNN model empowered by LLM for precise and efficient reasoning on KG, extracting high-quality reasoning chains. We further propose a knowledge-enhanced multiple-choice prompt to guide the LLM to reason for final answer.

- Extensive experiments on KGQA show that our method effectively combines GNN's structured, explicit learning with LLM's powerful language understanding, outperforming state-of-the-art methods while maintaining high efficiency and interpretability.

## 2. Related Work

**Dual-Process Theory.** Dual-process theory [20–22] is a psychological account of how human thinking and decision-making arise from two distinct systems. System 1, corresponding to the implicit process, is associative and intuitive, enabling quick comprehension through associations and pre-existing knowledge [7, 8]. System 2, corresponding to the explicit process, is more deliberate and logical. It operates on symbolic structures, conducting explicit reasoning to arrive at conclusions [8, 22]. These systems serve complementary functions and can collaborate for a reasoning problem [22].

**LLM reasoning.** LLMs [1–4], have demonstrated impressive capabilities in many tasks such as question answering by leveraging implicit, associative reasoning similar to the intuitive "System 1" process in dual-process theory [20, 22]. To further improve their performance, some approaches instruct LLMs to generate reasoning process in their outputs [5, 6, 26]. While these "System 2"-like deliberate and explicit reasoning methods can achieve better results, they typically entail higher inference costs [7]. Furthermore, LLMs may make mistakes during the reasoning process, due to limitations in their internal knowledge, leading to issues such as hallucinations and inaccuracies [9].

**KGQA.** Given a natural language question $q$ and a KG $\mathcal{G} = \{(e_s, r, e_o)|e_s, e_o \in \mathcal{V}, r \in \mathcal{R}\}$, where $\mathcal{V}$ is the set of entities (nodes) and $\mathcal{R}$ is the set of relation types, the task of *Question Answering over Knowledge Graph (KGQA)* is to find a function $\mathcal{F}(q, G)$ that predicts the answer entities $e_a \in \mathcal{V}$ of $q$ over KG $\mathcal{G}$. As a common and practical setting, for each question $q$, the involved topic entities $e_q \in \mathcal{V}$ and answer entities $e_a \in \mathcal{V}$ are both labeled in KG . KGQA plays a vital role in various intelligent systems, such as Apple Siri and Microsoft Cortana [27]. To solve this task, classical methods [28–31]

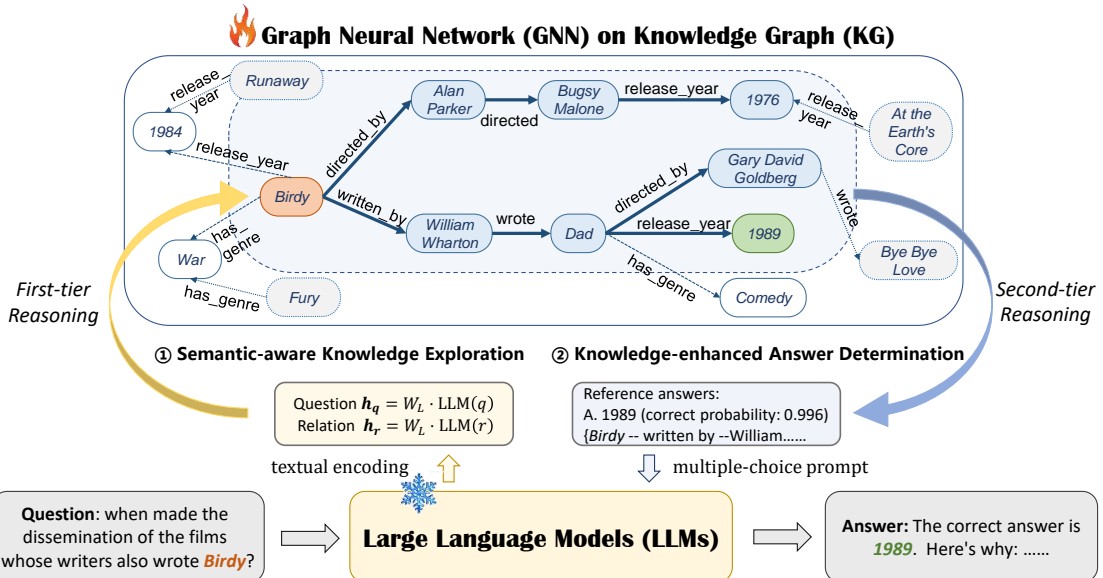

Figure 1: Illustration of the Dual-Reasoning (DualR), a GNN-LLM collaborative framework for knowledge graph question answering. It contains two-tier reasoning: (1) semantic-aware knowledge exploration; and (2) knowledge-enhanced answer determination.

usually first retrieve a question-related subgraph and then use different model (e.g., embedding model, graph neural networks[23, 24]) to reason for answers.

**LLM for KGQA.** Recently, considering the powerful language processing capabilities of LLMs, many works integrate LLMs with KG for KGQA task. These works typically follow the RAG paradigm [32, 33], which utilize various modules to extract knowledge from KG for LLMs generating final answers. Some methods retrieve information from KG based on text similarity [15, 34, 35]. For instance, KAPING [15] employs embedding model to retrieve triplets, but fails to leverage structured knowledge to do reasoning, resulting in the retrieval of excessive and irrelevant information [36]. Another strategy involves guiding the LLM itself to retrieve the knowledge from KG [17, 18, 37, 38]. For example, StructGPT [17] and ToG [18] view LLM as an agent, guiding it to search on the KG iteratively. But the depth and breadth of this search are limited as LLMs inherently lack the ability to comprehend graph structures [9]. Additionally, frequent interactions with LLMs are inflexible and entail high costs. RoG [38] fine-tunes an LLM to generate reasoning paths for information retrieval, but it requires substantial fine-tuning and may generate invalid path due to the hallucinations. A concurrent approach, GNN-RAG [39], utilizes an off-the-shelf GNN model to retrieve the shortest path to high-scoring candidate entities for LLMs. However, it depends on heuristic technique for path retrieval, which may cause low-quality reasoning chains. Moreover, supplying only the path does not fully leverage the GNN model, limiting its effectiveness. Notably, there is an independent category of methods, semantic parsing methods, which use LLMs to generate query languages and then execute them on KGs to obtain answers [40–43]. These methods typically rely on SPARQL query and are orthogonal to ours.

## 3. Proposed Method

### 3.1. Overview

Constructing high-quality reasoning chains is crucial for LLMs to generate correct answers. However, in the KGQA task, LLMs face challenges in effectively and efficiently extracting meaningful reasoning chains relevant to a given question from large and complex KGs. Following dual-process theory that describes human thinking as arising from two complementary systems, we propose a novel Dual-Reasoning (DualR) framework, which combines the strengths of GNN-based "System 2" and LLM-based "System 1" to achieve accurate, efficient and faithful reasoning over KGs.

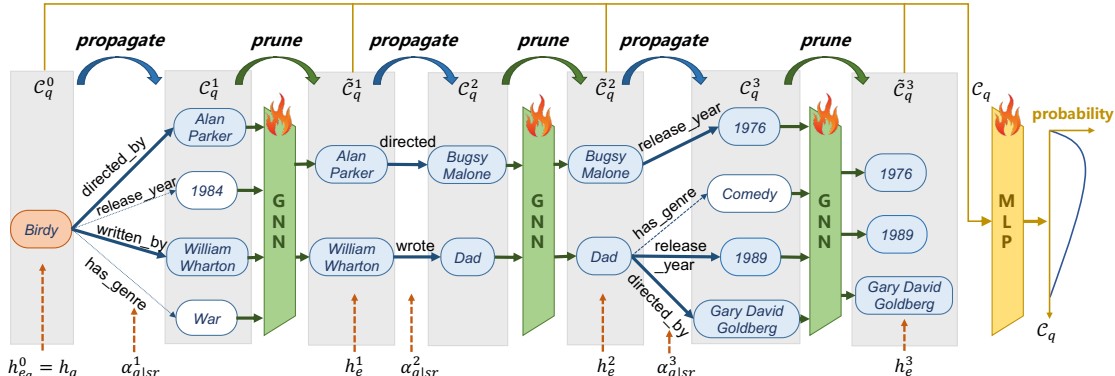

Figure 2: Illustration of semantic-aware knowledge exploration. We start exploration from topic entity *Birdy*. In each step, we firstly get the unpruned candidate set $\mathcal{C}_q^\ell$, calculate the attention weights $\alpha_{q|sr}^\ell$ of different edges, prune several irrelevant entities (in white), and update candidate set $\widetilde{\mathcal{C}}_q^\ell$ (in blue). The representations are propagated from entities in $\widetilde{\mathcal{C}}_q^{\ell-1}$ to $\widetilde{\mathcal{C}}_q^\ell$ through an one-layer GNN.

The proposed framework is shown in Figure 1, which contains two-tier reasoning: (1) semantic-aware knowledge exploration; and (2) knowledge-enhanced answer determination. For the first part, to implement the "System 2" efficiently, we design an LLM-empowered GNN module to explicitly reason on the graph, adaptively exploring fine-grained structured knowledge to construct meaningful reasoning chains. For the second part, to effectively leverage the extracted reasoning chains, we carefully design a knowledge-enhanced multiple-choice prompt, guiding the LLM to perform thoughtful reasoning, achieving accurate and reliable answer determination.

## 3.2. Semantic-Aware Knowledge Exploration

In this section, we aim to construct a deliberate, structured "System 2"-like module on the KG. Through explicit learning and reasoning, it explores structured knowledge on the graph, extracting meaningful reasoning chains that connect topic entity with promising candidates. Due to a large amount of irrelevant information present in the KG, we design an adaptive exploration GNN module that, with the semantic representation capabilities provided by the LLM, can automatically prune irrelevant information, enabling precise, efficient and interpretable reasoning on KG.

**Semantic-aware pruning.** Given the topic entity $e_q$ involved in the question $q$, we aim to gradually explore the potential answer entities starting from $e_q$ on the KG. Instead of directly using LLM to explore the candidates, we design a lightweight neural network empowered by LLM. Specifically, we initialize the candidate set $\mathcal{C}_q^0 \equiv \{e_q\}$. The representation of $e_q$ is initialized as the question encoding, i.e., $\boldsymbol{h}_{e_q}^0 = \boldsymbol{h}_q = \boldsymbol{W}_L \cdot \text{LLM}(q)$, where $\text{LLM}(\cdot)$ computes the average embedding in the first and last layer of Llama2-13B[3], and $\boldsymbol{W}_L \in \mathbb{R}^{d \times d_L}$ is a learnable weighting matrix mapping the representation to a lower dimension $d$. The representations of other entities are initialized as $\boldsymbol{0}$. Assume in the $\ell$-th step ($\ell = 1, 2, \ldots, L$), we have explored a set $\mathcal{C}_q^{\ell-1}$ of current candidates. The set is then expanded by propagating to the neighbors of entities in $\mathcal{C}_q^{\ell-1}$, resulting in an updated set $\mathcal{C}_q^\ell = \{e_o : (e_s, r, e_o) \in \mathcal{G}, e_s \in \mathcal{C}_q^{\ell-1}\}$.

Since KG contains a lot of information irrelevant to the question, we should filter the irrelevant edges during exploring such that the size of $\mathcal{C}_q^\ell$ will not grow exponentially. In the $\ell$-th step, we calculate the attention weight $\alpha_{q|sr}^\ell$, to measure the importance of each edge $(e_s, r, e_o)$ with $e_s \in \mathcal{C}_q^{\ell-1}$ as:

$$\alpha_{q|sr}^\ell = \sigma \left( \boldsymbol{W}_s^\ell \boldsymbol{h}_s^{\ell-1} + \boldsymbol{W}_r^\ell \boldsymbol{h}_r + \boldsymbol{W}_q^\ell \boldsymbol{h}_q + \boldsymbol{W}_{qr}^\ell (\boldsymbol{h}_r \odot \boldsymbol{h}_q) \right), \tag{1}$$

where $\sigma$ is the sigmoid function, $\boldsymbol{W}^\ell$'s in $\mathbb{R}^{1 \times d}$ are learnable weight matrices, and $\odot$ is the Hadamard product of vectors. The representations $\boldsymbol{h}_q$ and $\boldsymbol{h}_r$ are textual encodings mapped from an LLM (i.e., Llama2-13B) such that the semantic relevance of question with the current edge can be measured. Note that the LLM used as text encoding will not be updated. The representation $\boldsymbol{h}_s^{\ell-1} \in \mathbb{R}^d$ of head entity $e_s$ contains the knowledge learned in the $(\ell-1)$-th step.

By incorporating the representations of question $q$, relations, and entities within the triplets, we can utilize the powerful semantic modeling capabilities of LLM. Meanwhile, setting different weight matrices in each step allows us to capture essential information from different sections of the question, thereby preserving the triplets that are semantically relevant to the question. For each head node $e_s$, we select the top-$K$ edges $(e_s, r, e_o)$ based on $\alpha_{q|sr}^\ell$ of different edges and prune the others, resulting in a smaller candidate set $\widetilde{\mathcal{C}}_q^\ell$. Here, $K$ is a hyperparameter based on the characteristics of the KG. For example, Figure 2 illustrates a simple example with $K = 2$ and $L = 3$, where blue entities are retained while white entities irrelevant to the question are pruned. This way, we can adaptively explore the related information on the graph while reducing computation costs.

**GNN encoding through propagation.** To learn representations of each entity $e_o \in \widetilde{\mathcal{C}}_q^\ell$, we use a lightweight network, i.e., a 1-layer GNN, to propagate the information from entities $e_s \in \widetilde{\mathcal{C}}_q^{\ell-1}$ one step further to entities $e_o$ with

$$\boldsymbol{h}_o^\ell = \delta\Big(\sum\nolimits_{(e_s, r, e_o) \in \widetilde{\mathcal{N}}_{e_o}^\ell} \alpha_{q|sr}^\ell \boldsymbol{W}^\ell (\boldsymbol{h}_s^{\ell-1} \odot \boldsymbol{h}_r)\Big), \tag{2}$$

where $\delta(\cdot)$ is the activation function, $\widetilde{\mathcal{N}}_{e_o}^\ell$ is the set of preserved neighbor edges of tail entity $e_o$, $\boldsymbol{W}^\ell \in \mathbb{R}^{d \times d}$ is a learnable weight matrix in the $\ell$-th step, and the attention weight $\alpha_{q|sr}^\ell$ is computed in (1). In this way, the compositional information of edges relevant to question $q$ connecting from topic entity $e_q$ to $e_o$ can be propagated into $\boldsymbol{h}_o^\ell$.

As shown in Figure 2, after $L$ steps of propagation, we can form the final candidate set $\mathcal{C}_q = \widetilde{\mathcal{C}}_q^0 \cup \cdots \cup \widetilde{\mathcal{C}}_q^L$ and obtain their representations $\boldsymbol{h}_e^L$. Finally, we use a multi-layer perceptron (MLP) and softmax function on the entity representation $\boldsymbol{h}_{e_i}^L$ and the question representation $\boldsymbol{h}_q$ to obtain the probability of entity $e_i$ being the correct answer:

$$p(q, e_i) = e^{\mathsf{MLP}([\boldsymbol{h}_{e_i}^L; \boldsymbol{h}_q])} \Big/ \sum\nolimits_{\forall e_j \in \mathcal{C}_q} e^{\mathsf{MLP}([\boldsymbol{h}_{e_j}^L; \boldsymbol{h}_q])}. \tag{3}$$

We optimize the neural network with supervision given by the question-answer pairs. Specifically, we use the CrossEntropy Loss [44]:

$$\mathcal{L} = \sum\nolimits_{(q, e_a) \in \mathcal{F}_{tra}} - \log(p(q, e_a)), \tag{4}$$

where $\mathcal{F}_{tra}$ is the training set of question-answer pairs. The set of model parameters are randomly initialized and optimized by minimizing $\mathcal{L}$ with Adam stochastic gradient descent algorithm [45].

Through adaptive propagation with GNN, the "System 2"-like explicit reasoning we build on the KG can effectively explore structured knowledge, identifying promising candidate answers. Furthermore, by leveraging the weights of different edges, we can extract paths connecting the topic entity and candidate answers to construct meaningful reasoning chains, which will be detailed in Section 3.3.

### 3.3. Knowledge-Enhanced Answer Determination

Although the GNN module can indicate the probabilities of candidates from the output scores, leveraging the powerful language understanding capabilities of the LLM for aligning the question linguistically with the explored information can provide greater benefits. Therefore, we extract high-quality reasoning chains to guide the LLM in reasoning for final answer determination. Specifically, to enhance the associative "System 1"-like reasoning of LLMs, we design a knowledge-enhanced multiple-choice prompt, guiding LLMs in effectively combining external explicit knowledge with its own internal implicit knowledge, and generating answers with a rapid one-step inference.

**Reasoning chain extraction.** To extract valuable reasoning chains for LLM, we employ a greedy algorithm to trace back paths. Specifically, we first preserve top-$N$ candidate entities in $\mathcal{C}_q$ identified with the highest probabilities as reference answers. Then, we backtrack from each candidate answer $e_c$, select the edge from $\widetilde{\mathcal{N}}_{e_c}^L$ with the highest attention weight (as per (1)), and set the head entity as a new starting point in the next step. By conducting $L$-steps backtracking, it will eventually trace back to the initial topic entity $e_q$ and obtain a reasoning evidence chain between $e_q$ and $e_c$ (the detailed

algorithm is shown in Appendix B.3). These chains reveal the compositional associations between topic entities and candidate answers, providing faithful evidence for the final answer determination. For example, in Figure 1, the evidence chain connecting the topic entity *Birdy* and the reference answer *1989* is: $Birdy \xrightarrow{written\_by}$ William Wharton $\xrightarrow{wrote}$ Dad $\xrightarrow{release\_year}$ *1989*. In this way, LLMs can conduct semantic analysis and reasoning based on reasoning chains. For instance, it can recognize that the second reference answer (*1976*) in Figure 1 does not align with the semantic context of the question based on the chains between *Birdy* and *1976*.

**Knowledge-enhanced multiple-choice prompt.** To effectively leverage the extracted reasoning chains, we further refine them into a knowledge-enhanced multiple-choice prompt. We incorporate two additional types of knowledge to enhance the LLM's resoning for answer determination. First, we extract the list of candidate entity and assign them labels. This approach provides clear targets, effectively stimulating the LLM's associative and intuitive reasoning capabilities. Second, we use the correct probability (as per (3)) of each candidate answer to provide the confidence returned by the GNN module for LLM, which aids the LLM in engaging in reasoning with heightened attention.

Overall, the knowledge-enhanced multiple-choice prompt is built using a task description, the provided question, and reference answers, which include candidate entity, correct probability and reasoning chain, as shown on the right.

```
<Task Description>
<Question>
<Reference Answers>:
A. candidate 1 (correct probability) {reasoning chain}
B. candidate 2 (correct probability) {reasoning chain} ......
```

The complete input-output examples of the final answer determination are provided in the Appendix D, which demonstrates our method's capability for accurate and interpretable reasoning. In addition, to avoid distractions from irrelevant information, we do not offer any few-shot examples, providing a clear structure for the LLM to follow, while maintaining the benefits of zero-shot learning.

### 3.4. Training Strategy

During training, we freeze the LLM in both text encoding and answer determination steps to avoid expensive costs. Instead, we only update the lightweight GNN module by minimizing the loss in (4). Moreover, unlike traditional GNN-based approaches [29–31], which first retrieve a subgraph and then perform reasoning on it, our GNN module can unify and simultaneously perform these two steps, adaptively filter out irrelevant information on the graph, and achieve more efficient reasoning. Additionally, it is noteworthy that the answer determination module can be adapted to any pre-trained LLM with just one-step inference, without the need for fine-tuning or frequent interaction, thereby avoiding time and resource overhead. In this way, the GNN module can served as "System 2", as it performs explicit, step-by-step reasoning on graph structures, while the LLM aligns with "System 1", leveraging implicit, associative patterns for rapid and intuitive decision-making.

Considering there are common compositional relationships between questions and the concepts in the graphs, the GNN module can be further benefited from pre-training strategies. Specifically, we pre-train the networks on two comprehensive KGQA datasets WebQSP [46] and CWQ [47], and then fine-tune them on target datasets. This approach enables the GNN module to better learn the compositional relationships, enhancing question understanding and generalization abilities.

## 4. Experiments

### 4.1. Experimental Setup

**Datasets.** Following existing KGQA works [30, 31], we use three benchmark datasets, namely WebQSP [46], CWQ [47], and MetaQA [48], to evaluate different methods. The MetaQA dataset is divided into three versions based on the number of hops required in KG, namely 1-hop, 2-hop, and 3-hop. Table 1 displays the statistics of these three datasets.

Table 1: Statistics of KGQA datasets.

| Datasets | #Train | #Valid | #Test | Max #hop |
|---|---|---|---|---|
| WebQSP | 2,848 | 250 | 1,639 | 2 |
| CWQ | 27,639 | 3,519 | 3,531 | 4 |
| MetaQA-1 | 96,106 | 9,990 | 9,947 | 1 |
| MetaQA-2 | 118,980 | 14,872 | 14,872 | 2 |
| MetaQA-3 | 114,196 | 14,274 | 14,274 | 3 |

Table 2: Performance comparison of different methods for KGQA (Hits@1 in percent).

| Type | Methods | WebQSP | CWQ | MetaQA-1 | MetaQA-2 | MetaQA-3 |
|---|---|---|---|---|---|---|
| KG-based | KV-Mem | 46.7 | 18.4 | 96.2 | 82.7 | 48.9 |
| | GraftNet | 66.4 | 36.8 | 97.0 | 94.8 | 77.7 |
| | EmbedKGQA | 66.6 | - | 97.5 | 98.8 | 94.8 |
| | NSM | 68.7 | 47.6 | 97.1 | **99.9** | 98.9 |
| | SR+NSM | 69.5 | 50.2 | - | - | - |
| | UniKGQA | 75.1 | 50.7 | 97.5 | 99.0 | 99.1 |
| GPT-3-based | KB-Binder | 74.4 | - | 92.9 | **99.9** | 99.5 |
| | KAPING | 73.9 | 55.4 | - | - | - |
| Llama2-based | Llama2-13B | 40.9 | 22.1 | 31.9 | 15.8 | 34.9 |
| | RoG-Llama2-7B | 74.2 | 56.4 | - | - | - |
| | ToG-Llama2-70B | 68.9 | 57.6 | - | - | - |
| | **DualR-Llama2-13B** | **78.3** | **58.0** | **97.9** | **99.1** | **99.6** |
| ChatGPT-based | ChatGPT | 61.2 | 38.8 | 61.9 | 31.0 | 43.2 |
| | RoG-ChatGPT | 81.5 | 52.7 | - | - | - |
| | KD-CoT | 68.6 | 55.7 | - | - | - |
| | StructGPT | 72.6 | 55.3 | 94.2 | 93.9 | 80.2 |
| | ToG-ChatGPT | 76.2 | 58.9 | - | - | - |
| | **DualR-ChatGPT** | **82.8** | **62.0** | **98.1** | **99.7** | **99.7** |
| GPT-4-based | GPT-4 | 67.3 | 46.0 | 65.7 | 34.6 | 48.9 |
| | ToG-GPT-4 | 82.6 | 69.5 | - | - | - |
| | **DualR-GPT-4** | **87.6** | **73.6** | **98.3** | **99.9** | **99.9** |

**Evaluation Metrics.** Following [17, 18], we focus on generating the answer with the highest confidence, and use Hits@1 to evaluate whether the top-ranked predicted answer is correct.

**Experiment Details.** In the pre-training stage, we set the dimension $d$ as 256 for the GNN module, learning rate as 1e-4, batch size as 20, number of layers $L$ as 3 and number of sampling $K$ as 200. As for the fine-tuning stage, we adjust the $L$ and $K$ based on the performance on validation set, and details are described in Appendix B.4. Considering the plug-and-play convenience of DualR, we use three LLMs for answer determination in experiments: Llama2-13B-chat [3], ChatGPT and GPT-4[2]. We typically set number of reference answers $N$ as 3, and the influence of $N$ is shown in Appendix C.4. The GNN module is trained on an RTX 3090-24GB GPU, while inference for Llama2-13B-chat runs on two RTX 3090-24GB GPUs.

**Baseline Methods.** We consider following baseline methods for performance comparison: (1) KG-based methods without using LLMs: KV-Mem [49], GraftNet [29], EmbedKGQA [28], NSM [30], SR+NSM [50], UniKGQA [31]; (2) LLM-based methods: KB-Binder [42] based on Codex [51], KAPING [15] based on GPT-3 [1], RoG [38] that can be plug-and-play with different LLMs, KD-CoT [37] and StructGPT [17] based on ChatGPT, ToG [18] that can be plug-and-play with different LLMs.

## 4.2. Performance Comparison

**Main results.** From the results in Table 2, it can be observed that our method DualR, whether combined with the Llama2 or GPT, outperforms traditional methods without LLMs. From the last three groups of the table, it can be seen that incorporating KG can effectively enhance the performance of LLMs, as they often lack the knowledge relevant to the questions. Within methods using Llama2, DualR combined with the 13B model outperforms ToG which is combined with the 70B model, and also outperforms some methods using ChatGPT. Similarly, within methods using ChatGPT and GPT-4, our approach demonstrates significant advantages. This shows the superiority of proposed dual-reasoning framework, effectively harnessing the GNN's precise explicit learning and the LLM's powerful language understanding, outperforming LLM's single reasoning significantly. Additionally, the performance of DualR improves with the integration of more powerful LLM. This demonstrates that DualR effectively combines explicit knowledge from KGs with implicit knowledge from LLM, leading to more accurate reasoning.

---

[2] `https://openai.com/`

Table 3: Comparison of inference time (seconds) and number of interaction with LLM (Llama2-13B) per question of different methods.

| Methods | WebQSP | | CWQ | |
|---|---|---|---|---|
| | time | #interaction | time | #interaction |
| RoG | 1.98 | 2 | 3.04 | 2 |
| StructGPT | 3.37 | 3 | 4.22 | 4 |
| ToG | 16.7 | 15 | 20.5 | 22 |
| DualR | **1.29** | **1** | **1.99** | **1** |

Table 4: Comparison of different variants of DualR-Llama2-13B in Hits@1(%).

| Methods | WebQSP | CWQ |
|---|---|---|
| DualR | 78.3 | 58.0 |
| w.o.-mcp | 72.9 | 54.1 |
| w.o.-cand | 77.5 | 52.5 |
| w.o.-prob | 76.2 | 55.4 |
| w.o.-chain | 76.8 | 56.0 |

**Efficiency comparison.** We also compare the inference time and number of interaction with LLM of several representative LLM-based methods, executing Llama2-13B within our local environment. As can be seen in Table 3, RoG requires two interactions with LLM, while step-by-step reasoning methods StructGPT and ToG require even more frequent interactions, thus incurring a high cost. However, in our approach, question encoding of LLM and graph reasoning of GNN are highly efficient, and we ultimately requires only one-step inference of LLM, thereby achieving high efficiency.

**Fine-tuned setting results.** While our method can seamlessly integrate with any off-the-shelf LLM, we expect that instruction fine-tuning the LLM for answer determination can lead to further performance improvements. We compare our approach with two recent state-of-the-art methods, RoG and GNN-RAG [39], both of which fine-tune Llama2-7B-chat on WebQSP and CWQ datasets, using Hit, Hits@1 and F1 as evaluation metrics. As shown in Table 5, DualR achieves state-of-the-art performance across multiple metrics, especially showing greater gains on more challenging dataset (i.e., CWQ), further validating the effectiveness of our method.

Table 5: Performance comparison of different methods with fine-tuned Llama2-7B-chat.

| Methods | WebQSP | | | CWQ | | |
|---|---|---|---|---|---|---|
| | Hit | Hits@1 | F1 | Hit | Hits@1 | F1 |
| RoG | **85.7** | 80.0 | 70.8 | 62.6 | 57.8 | 56.2 |
| GNN-RAG | **85.7** | 80.6 | 71.3 | 66.8 | 61.7 | 59.4 |
| DualR | 84.9 | **81.5** | **71.6** | **68.9** | **65.3** | **62.1** |

## 4.3. Ablation Study

### 4.3.1. Effectiveness of Knowledge Exploration

In this section, we analyze the effectiveness of our GNN module, focusing on the Hits@1 of its independent output answer without the answer determination by LLM (i.e., DualR-w.o.-AD). Here, we choose UniKGQA, the state-of-the-art KG-based model without LLMs for comparison. As can be seen in Table 6, our GNN module outperforms UniKGQA on all datasets, demonstrating the effectiveness of the designed network for topology and semantic aware reasoning on the graph, capable of identifying promising candidates and extracting high-quality reasoning chains.

### 4.3.2. Effectiveness of LLM's Answer Determination

Furthermore, Table 6 demonstrates the effectiveness of using LLM to reason for answer determination. Although the GNN module itself (DualR-w.o.-AD) achieves decent performance, leveraging extracted reasoning chains to guide LLM for answer determination brings significant gains. With its powerful language processing capabilities, the LLM can analyze the semantic relationships between a given question and relevant reasoning chains and make the thoughtful determination. By leveraging the implicit knowledge embedded within itself, it achieves more accurate reasoning.

### 4.3.3. Influence of Prompt Form for Answer Determination

We design different forms of prompts to evaluate their influence in guiding answer determination of DualR-Llama2-13B. The variants include: (1) not using multiple-choice prompt (DualR-w.o.-mcp), (2) not using candidate entities (DualR-w.o.-cand), (3) not using correct probabilities (DualR-w.o.-prob), and (4) not using reasoning chains (DualR-w.o.-chain), which are shown in Appendix E.

Table 6: Comparison of Different Variants of DualR in Hits@1(%).

| Methods | WebQSP | CWQ | MetaQA-1 | MetaQA-2 | MetaQA-3 |
|---|---|---|---|---|---|
| UniKGQA | 75.1 | 50.7 | 97.5 | 99.0 | 99.1 |
| DualR-w.o.-AD | 76.8 | 55.6 | 97.6 | 99.1 | 99.5 |
| DualR-Llama2-13B | 78.3 | 58.0 | 97.9 | 99.1 | 99.6 |
| DualR-ChatGPT | 82.8 | 62.0 | 98.1 | 99.7 | 99.7 |

As shown in Table 4, compared with DualR-w.o.-mcp, which inputs all candidate answers continuously without specific labels, our designed multiple-choice prompt can achieve better performance, since it offers a clearer structure and strength the connection of candidate answers with reasoning chains. Moreover, all three factors in the prompt, i.e., candidate entity, correct probability and reasoning chain, have a positive impact on guiding LLM to make correct choices. Providing candidate answers can refine the target for LLM (compared with DualR-w.o.-cand), and correct probabilities prompts LLM to engage in reasoning with attention (compared with DualR-w.o.-prob). Reasoning paths are crucial as they provide detailed process for LLM reasoning (compared with DualR-w.o.-chain), enhancing the accuracy and reliability of the output, as can be seen in Section 4.4.

## 4.4. Case Study

We present two case studies in Figure 3, which display the question, reference answers of the input prompt, and the final result of the LLM's answer determination. The task description can be found in Appendix D. It can be observed that the LLM without Dual-Reasoning (i.e., Llama2-13B-CoT) may suffer from hallucination and lack of knowledge, resulting in wrong answers. In contrast, DualR-Llama2-13B generates the correct answers, even when the right answer is not ranked at the top. Moreover, based on the given information, the LLM provides its reasoning process, which is consist with the input reasoning chains, enhancing the credibility of the output. These examples illustrate how DualR effectively implement LLM's intuitive reasoning through externalized reasoning chains, achieving accurate and faithful results.

(a) One example from MetaQA-3hop.  (b) One example from CWQ.

Figure 3: Examples of accurate and faithful output by DualR-Llama2-13B.

## 5. Conclusion

In this paper, we introduce Dual-Reasoning (DualR), a novel framework following dual-process theory, which combines GNN-based structured, deliberate reasoning of "System 2" with LLM-based intuitive reasoning of "System 1" in KGQA task. DualR designs an LLM-empowered GNN module to explicitly reason on KGs, extracting high-quality reasoning chains, which are then refined into a knowledge-enhanced multiple-choice prompt, guiding the LLM to reason for final answer. With the collaboration of GNN and LLM, this two-tier reasoning process harnesses the precision graph learning of GNN and the language understanding prowess of LLM. Extensive experiments demonstrate the superiority of DualR, achieving accurate, efficient and faithful reasoning over KGs.

# Acknowledgements

This work is supported by National Key Research and Development Program of China (under Grant No.2023YFB2903904), National Natural Science Foundation of China (under Grant No.92270106) and Beijing Natural Science Foundation (under Grant No.4242039).

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

# A. Discussion

In our problem setting, the answers are restricted to entities within the KG, which might be too strict in real-world applications. However, as long as the answer can be linked to entities within the KG, we can still obtain supervision signals to train the GNN model and complete the entire process. Specifically, we can also leverage LLMs to generate natural questions based on the multi-hop relations between entities, and incorporating supervision signals from intermediate entities to further enhance model training. This presents a promising direction for future research.

# B. Implementation Details

## B.1. Question and Relation Encoding

Considering the remarkable modeling capacity of the LLM, we first employ the average embedding in the first and last layer of the LLM (i.e., Llama2-13B) to produce text encoding as the representations of question $q$ and relation $r$ in the KG:

$$\bar{\boldsymbol{h}}_q = \text{LLM}(q), \bar{\boldsymbol{h}}_r = \text{LLM}(r). \tag{5}$$

Note that the parameter dimension $d_L$ of the LLM is typically high, so we use a weight matrix $\boldsymbol{W}_L$ to reduce the dimension from $d_L$ to $d$, i.e., $\boldsymbol{h}_q = \boldsymbol{W}_L \bar{\boldsymbol{h}}_q, \boldsymbol{h}_r = \boldsymbol{W}_L \bar{\boldsymbol{h}}_r$.

It is worth noting that the reverse relations $(-r)$ play an important role in graph reasoning, but there is no golden rule for obtaining the textual representation of them. So we use a linear layer to map $\boldsymbol{h}_r$, generating the representation $\boldsymbol{h}_{-r}$ of reverse relation $-r$, i.e., $\boldsymbol{h}_{-r} = \boldsymbol{W}_{-r} \boldsymbol{h}_r + \boldsymbol{b}_{-r}$. Additionally, we separately learn a representation $\boldsymbol{h}_{id}$ for the identity relation. In this way, we can leverage LLM's capabilities to achieve unified representation, effectively mining semantic information from the KG.

## B.2. Exploration Algorithm

---

**Algorithm 1** Semantic-aware knowledge exploration.

---

**Require:**
   question $q$, topic entity $e_q$, KG $G$, question encoding $\boldsymbol{h}_q$, relations encoding $\boldsymbol{h}_r$'s, depth $L$, model parameters $\boldsymbol{\Theta}$.
1: initialize $\boldsymbol{h}_{e_q}^0 = \boldsymbol{h}_q$ and $\widetilde{\mathcal{C}}_q^0 = \{e_q\}$;
2: **for** $l = 1, 2 \cdots L$ **do**
3:    get the candidate set $\mathcal{C}_q^\ell = \{e_o : (e_s, r, e_o) \in \mathcal{G}, e_s \in \widetilde{\mathcal{C}}_q^{\ell-1}\}$
4:    calculate attention weights $\alpha_{q|sr}^\ell$ (by (1));
5:    get the pruned candidate set $\widetilde{\mathcal{C}}_q^\ell$ based on different $\alpha_{q|sr}^\ell$;
6:    **for** $e \in \widetilde{\mathcal{C}}_q^\ell$ (in parallel) **do**
7:       $\boldsymbol{h}_o^\ell := \delta(\sum_{(e_s,r,e_o) \in \widetilde{\mathcal{N}}_{e_o}^\ell} \alpha_{q|sr}^\ell \boldsymbol{W}^\ell (\boldsymbol{h}_s^{\ell-1} \odot \boldsymbol{h}_r))$ (by (2)).
8:    **end for**
9: **end for**
10: **return** $\boldsymbol{h}_e^L$ for all $e \in \mathcal{C}_q = \widetilde{\mathcal{C}}_q^0 \cup \cdots \cup \widetilde{\mathcal{C}}_q^L$.

---

## B.3. Path Backtracking Algorithm

---

**Algorithm 2** Path Backtracking.

---

**Require:**
    question $q$, KG $\mathcal{G}$, depth $L$, model parameters $\Theta$, hyperparameter $N$.
1: run Algorithm 1, obtain attention weights $\{\alpha^\ell_{q|sr}\}_{\ell=1,\ldots,L}$ of different edges in each step $\ell$;
2: select top-$N$ candidates entities to form $\widetilde{\mathcal{C}}_q$
3: **for** $e_c \in \widetilde{\mathcal{C}}_q$ **do**
4:    set $e_t = e_c$;
5:    **for** $l = L, L-1 \cdots 1$ **do**
6:        obtain attention weights $\{\alpha^\ell_{q|sr}\}$ of edges whose tail node is $e_t$;
7:        $\mathcal{E}^\ell_{q|e_c} = \{(e_h, r, e_t) : h = \arg\max_s \alpha^\ell_{q|sr}\}$;
8:        $e_t = e_h$;
9:    **end for**
10: **end for**
11: **return** $\mathcal{P}_{q|e_c} = \mathcal{E}^1_{q|e_c} \cup \cdots \cup \mathcal{E}^L_{q|e_c}$ for $e_c \in \widetilde{\mathcal{C}}_q$ .

---

## B.4. Details of Experiments.

**Datasets.** We adopt three benchmark KGQA datasets: WebQuestionSP (WebQSP)[46], Complex WebQuestions (CWQ) [47] and MetaQA [48] in this work. For WebQSP and CWQ, the corresponding KGs are Freebase [52]. Following previous works [30], we reduce the size of Freebase by extracting all triples that contain within the respective max reasoning hops of the topic entities for each question. For MetaQA, we directly use the original WikiMovies KG [3]. The statistics of three KGs are presented in Table 7.

Table 7: Statistics of three knowledge graphs.

| KG | #Entities | #Relations | #Triples |
|---|---|---|---|
| Freebase-WebQSP | 1,441,421 | 6,102 | 20,111,715 |
| Freebase-CWQ | 2,429,346 | 6,649 | 138,785,703 |
| WikiMovies-MetaQA | 43,234 | 9 | 134,741 |

**Hyperparameters.** During the pre-training stage of exploration module, (with the maximum number of training epochs set to 30), we set the learning rate as 1e-4, weight decay as 1e-3, batch size as 20, dimension $d$ as 256, number of layers $L$ as 3 and number of sampling $K$ as 200. As for the fine-tuning stage, we tune the learning rate in $[10^{-6}, 10^{-3}]$, weight decay in $[10^{-5}, 10^{-2}]$. We also adjust the $L$ and $K$ based on the performance on validation set, which is shown in Table 8.

Table 8: Hyperparameters of exploration module on different datasets.

| | WebQSP | CWQ | MetaQA-1 | MetaQA-2 | MetaQA-3 |
|---|---|---|---|---|---|
| $L$ | 2 | 4 | 1 | 2 | 3 |
| $K$ | 200 | 200 | 40 | 60 | 100 |

## B.5. Baselines

We consider the following baseline methods for performance comparison:

(1) traditional KG-based methods without using LLMs:

---

[3] https://research.fb.com/downloads/babi

- KV-Mem [49] utilizes a Key-Value memory network to store triples and conduct iterative read operations to deduce the answer.
- GraftNet [29] first retrieves the question-relevant triplets and text sentences from the KG and corpus to build a heterogeneous subgraph. Then it adopts a graph neural network to perform multi-hop reasoning on the subgraph.
- EmbedKGQA [28] embeds entities on KG and design a scoring function to rank them based on their relevance to the question.
- NSM [30], first conducts subgraph retrieval and then employ the neural state machine with a teacher-student network for multi-hop reasoning on the KG.
- SR+NSM [50] first employs a pretrained language model to build a subgraph retriever, then use NSM for reasoning on the retrieved subgraph.
- UniKGQA [31] conducts the subgraph retrieval and reasoning process with the same model based on GNN, which achieve salient performance on KGQA task.

(2) LLM-based methods:

- KB-Binder [42] first uses Codex [51] to generates logical forms as the draft by imitating a few demonstrations. Then it bind the generated draft to an executable one with BM25 score matching.
- KAPING [15] uses a sentence embedding model to retrieve the relevant triplets to the question from KG which are then forwarded to LLMs to generate the answer.
- RoG [38] first fine-tunes the Llama2-7B to generate relation paths grounded by KGs as faithful plans. Then it uses these plans to retrieve reasoning paths from the KGs for LLMs to conduct reasoning.
- KD-CoT [37] utilizes a QA system to verify and modify reasoning traces in CoT of LLMs via interaction with external knowledge source.
- StructGPT [17] views LLMs as agents and establishes an information interaction mechanism between KG and LLMs to iteratively deduce the answer to the question.
- ToG [18] improves StructGPT by guiding the LLM to iteratively execute beam search on KG.
- GNN-RAG [39] utilizes an off-the-shelf GNN model to retrieve the shortest path to high-scoring candidate entities for LLMs to reason for answers.

In Table 9, we summarize the differences between our method DualR and several representative baselines in KGQA. These methods generally involve knowledge retrieval and answer generation processes, and our knowledge exploration process can be seen as a form of knowledge retrieval. As can be seen, our method uniquely synergizes LLM with GNN, harnessing both the precision of graph learning and the prowess of language understanding.

Another category of methods, such as QA-GNN[53] and GreaseLM[54], integrates the GNN with the language model at the architectural level, making them incompatible with current LLMs. Additionally, their approach is specifically designed for multiple-choice question answering, where the model input requires candidate answers, and they lack the capability to search for candidates over the entire knowledge graph. Therefore, they are unable to complete the task in this paper.

## C. Supplementary Experiments

### C.1. Influence of Pruning

In this section, we discuss the influence of the pruning for candidate set. As shown in Figure 4, the size of the unfiltered candidate set is typically vast due to the large-scale knowledge graph. In contrast, after pruning, the candidate set is significantly reduced, which greatly decreases the computational cost during the GNN propagation process. Therefore, we can filter out a large amount of irrelevant information, significantly shortening inference time and achieving more efficient reasoning.

Table 9: Comparison of different methods.

| Methods | Knowledge Retrieval | Answer Generation |
|---------|---------------------|-------------------|
| UniKGQA | GNN | GNN |
| KAPING | sentence embedding model | LLM |
| RoG | fine-tuned LLM | LLM |
| StructGPT | LLM | LLM |
| ToG | LLM | LLM |
| GNN-RAG | GNN | LLM |
| DualR | GNN+LLM | LLM |

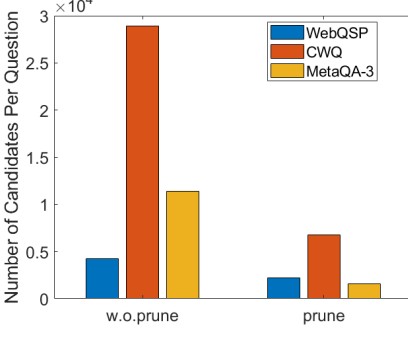
(a) Number of candidates.

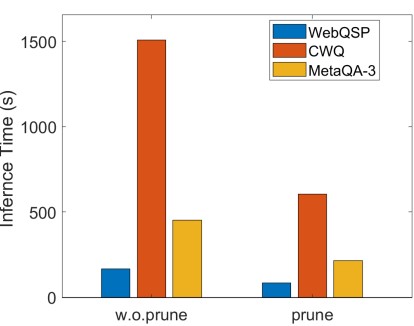
(b) Inference time.

Figure 4: Influence of pruning on three datasets.

## C.2. Effect of Pre-training Strategy

Table 10: Influence of pretraining of exploration module.

| Methods | WebQSP | CWQ | MetaQA-1 | MetaQA-2 | MetaQA-3 |
|---------|--------|-----|----------|----------|----------|
| w.o.-pretrain | 72.9 | 53.4 | 97.8 | 99.0 | 99.4 |
| pretrain-finetune | 76.8 | 55.6 | 97.8 | 99.1 | 99.5 |

In Section 3.4, we have mentioned that the GNN module can be benefited from pre-training strategies. We compare the performance of individual training strategy and pretrain-finetune strategy in Table 10. It is evident that the strategy of pre-training significantly outperforms training separately on distinct datasets. This underscores the effectiveness of pre-training in enabling our exploration module to better grasp the common compositional relationships between questions and the concepts in the graphs, thereby enhancing the question-comprehension abilities empowered by LLM.

We plot the learning curves for both training strategies. As can be seen in Figure 5, pre-training significantly accelerates the convergence speed of the model during fine-tuning on the downstream datasets, and acquires better performance. This advantage is more pronounced on more complex datasets (i.e., MetaQA-3hop).

To further validate the generalization capability of our method, we also conduct fine-tuning experiments on a dataset from the sports domain. We use the dataset WorldCup2014[55], which contains about 8000 questions with answers related to the 2014 World Cup, and questions are a mixture of 1-hop and 2-hop questions. As shown in table 11, our method achieve an impressive result of 100% on Hits@1, which is also superior to the existing baselines.

## C.3. Influence of Question and Relation Encoder

In our GNN module, it is feasible to use a small language model (e.g., RoBERTa[58]) to encode the text of questions and relations, since the GNN module is capable of learning and adapting it. However, our experiments have shown that using Llama2 for encoding produces superior results, as

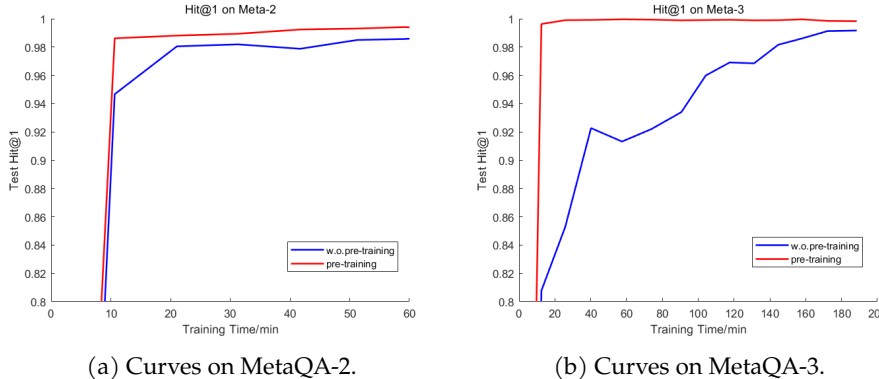

(a) Curves on MetaQA-2.  (b) Curves on MetaQA-3.

Figure 5: Learning curves on two datasets.

Table 11: Performance comparison on WorldCup2014 dataset.

| Methods | WC-1 | WC-2 | WC-m |
|---|---|---|---|
| TransferNet[56] | 97.9 | 96.5 | 96.8 |
| AlAgha [57] | 97.4 | 98.6 | 96.0 |
| DualR-Llama2-13B | 100 | 100 | 100 |

evidenced by the performance presented in the table 12. This improvement is attributed to Llama2's higher dimensionality and deeper processing in encoding, which allows it to capture semantic information and model text representations more effectively.

### C.4. Influence of Number of Reference Answer

Table 13 shows the impact of the number of reference answers in the determination module. As can be seen, providing three reference answers is appropriate, since more reference answers will introduce too much redundant information, interfering with the inference of the LLM.

## D. Case Study

Table 14 and 15 present examples of the input-output of our method DualR. Compared with Figure 3, we present the complete set of prompts, including the task description. It can be observed that by combining explicit knowledge from the KG with the implicit knowledge of LLM itself, LLM ultimately selects the correct answer and provides its reasoning process, enhancing the credibility of the output.

## E. Variants of Prompt

Table 17 shows the variants of designed prompt introduced in Section 4.3.3, Compared with these formats, our designed knowledge-enhanced multiple-choice prompt (in Table 16) can achieve better performance.

Table 12: Performance comparison of different encoders of DualR-w.o.-AD in Hits@1(%).

| Encoder | WebQSP | CWQ |
|---|---|---|
| RoBERTa-large | 75.2 | 53.9 |
| Llama2-13B | 76.8 | 55.6 |

Table 13: Comparison of different numbers of reference answers $N$ of DualR-Llama2-13B.

| N | 3 | 4 | 5 | 6 |
|---|---|---|---|---|
| WebQSP | 78.31 | 77.80 | 77.18 | 76.72 |
| CWQ | 58.02 | 57.87 | 57.43 | 56.55 |

Table 14: Case study 1 from MetaQA-3hop.

| | |
|---|---|
| Question–Answer | When made the dissemination of the films whose writers also wrote Birdy? - 1989. |
| Output by Llama2-13B-CoT | The films whose writers also wrote Birdy were disseminated in the following years: * The Right Stuff (1983) * The Big Red One (1980) Both of these films were written by Larry McMurtry and Daniel Taradash, who also co-wrote the screenplay for Birdy. |
| Input Task Description | Given a question, and the reference answers with their correct probabilities and associated retrieved knowledge graph triples (entity, relation, entity) as related facts, you are asked to answer the question with this information and your own knowledge. If the reference answers contain the correct answer, please output the label and content of the answer; If not, please answer the question based on your own knowledge. |
| Input Question and Reference Answers | Question: when made the dissemination of the films whose writers also wrote Birdy? Reference answers: A. 1989 (correct probability: 0.996) {relevant facts: (Birdy, written_by, William Wharton), (Dad, written_by, William Wharton), (Dad, release_year, 1989)} B. 1998 (correct probability: 0.004) {relevant facts: (Birdy, has_tags, nicolas cage), (Snake Eyes, has_tags, nicolas cage), (Snake Eyes, release_year, 1998)} C. 1976 (correct probability: 0.0) {relevant facts: (Birdy, directed_by, Alan Parker), (Bugsy Malone, directed_by, Alan Parker), (Bugsy Malone, release_year, 1976)} |
| Output by DualR-Llama2-13B | Based on the reference answers and my own knowledge, I can answer the question. The correct answer is A. 1989. Here's why: The reference answers provide three relevant facts: 1. (Birdy, written_by, William Wharton) 2. (Dad, written_by, William Wharton) 3. (Dad, release_year, 1989) Based on these facts, we can infer that the film "Dad" was written by William Wharton and released in 1989. Therefore, the correct answer is A. 1989. |

Table 15: Case study 2 from CWQ.

| | |
|---|---|
| Question-Answer | What is the most current year the team who won the 2010 World Series has won another World Series? - 2014 World Series. |
| Output by Llama2-13B-CoT | The team that won the 2010 World Series, the San Francisco Giants, has not won another World Series since then. Therefore, the most current year they have won a World Series is 2010. |
| Input Task Description | Given a question, and the reference answers with their correct probabilities and associated retrieved knowledge graph triples (entity, relation, entity) as related facts, you are asked to answer the question with this information and your own knowledge. If the reference answers contain the correct answer, please output the label and content of the answer; If not, please answer the question based on your own knowledge. |
| Input Question and Reference Answers | Question: What is the most current year the team who won the 2010 World Series has won another World Series?
Reference answers:
A. 2010 World Series (correct probability: 0.487) {relevant facts: (San Francisco Giants, time.participant.event, 2010 World Series), (San Francisco Giants, sports.sports_team.championships, 2010 World Series)}
B. 2014 World Series (correct probability: 0.26) {relevant facts: (San Francisco Giants, sports.sports_team.championships, 2010 World Series), (San Francisco Giants, sports.sports_team.championships, 2014 World Series)}
C. 2012 World Series (correct probability: 0.253) {relevant facts: (San Francisco Giants, time.participant.event, 2010 World Series), (San Francisco Giants, sports.sports_team.championships, 2012 World Series)} |
| Output by DualR-Llama2-13B | Based on the information provided, the most current year the team who won the 2010 World Series has won another World Series is: B. 2014 World Series. The relevant facts provided in the reference answer indicates that San Francisco Giants won 2010 World Series and 2014 World Series. Therefore, the answer is B. 2014 World Series. |

Table 16: Case study 3 from CWQ.

| Question-Answer | What's the home field for the sports team owned by Mark Attanasio? - Miller Park. |
|---|---|
| Input Task Description | Given a question, and the reference answers with their correct probabilities and associated retrieved knowledge graph triples (entity, relation, entity) as related facts, you are asked to answer the question with this information and your own knowledge. If the reference answers contain the correct answer, please output the label and content of the answer; If not, please answer the question based on your own knowledge. |
| Input Question and Reference Answers | Question: What's the home field for the sports team owned by Mark Attanasio?
Reference answers:
A. Seattle Pilots (correct probability: 0.7) {relevant facts: (Milwaukee Brewers, sports.professional_sports_team.owner_s, Mark Attanasio), (Milwaukee Brewers, sports.sports_team.previously_known_as, Seattle Pilots)}
B. Miller Park (correct probability: 0.285) {relevant facts: (Milwaukee Brewers, sports.professional_sports_team.owner_s, Mark Attanasio), (Milwaukee Brewers, sports.sports_team.arena_stadium, Miller Park)}
C. Milwaukee (correct probability: 0.015) {relevant facts: (Milwaukee Brewers, sports.professional_sports_team.owner_s, Mark Attanasio), (Milwaukee Brewers, sports.sports_team.location, Milwaukee)} |
| Output by DualR-Llama2-13B | Based on the information provided, the correct answer is B. Miller Park.
Here's why: The correct reference answer with the correct probability is B. Miller Park. The relevant facts provided are: * (Milwaukee Brewers, sports.professional_sports_team.owner_s, Mark Attanasio) * (Milwaukee Brewers, sports.sports_team.arena_stadium, Miller Park). These facts indicate that Miller Park is the home field of the Milwaukee Brewers, which is owned by Mark Attanasio. Therefore, the answer is B. Miller Park. |

Table 17: Different variants of knowledge-enhanced multiple-choice prompt.

| | prompt |
|---|---|
| DualR-w.o.-mcp | Reference answers include: [Seattle Pilots, Miller Park, Milwaukee]. Their correct probabilities are [0.7, 0.285, 0.015]. Relevant facts are [ (Milwaukee Brewers, sports.professional_sports_team.owner_s, Mark Attanasio), (Milwaukee Brewers, sports.sports_team.previously_known_as, Seattle Pilots), (Milwaukee Brewers, sports.sports_team.arena_stadium, Miller Park), (Milwaukee Brewers, sports.sports_team.location, Milwaukee)]. |
| DualR-w.o.-cand | Relevant facts include:
{(Milwaukee Brewers, sports.professional_sports_team.owner_s, Mark Attanasio), (Milwaukee Brewers, sports.sports_team.previously_known_as, Seattle Pilots)}(correct probability: 0.7)
{(Milwaukee Brewers, sports.professional_sports_team.owner_s, Mark Attanasio), (Milwaukee Brewers, sports.sports_team.arena_stadium, Miller Park)}(correct probability: 0.285)
{(Milwaukee Brewers, sports.professional_sports_team.owner_s, Mark Attanasio), (Milwaukee Brewers, sports.sports_team.location, Milwaukee)}(correct probability: 0.015). |
| DualR-w.o.-prob | Reference answers:
A. Seattle Pilots {relevant facts: (Milwaukee Brewers, sports.professional_sports_team.owner_s, Mark Attanasio), (Milwaukee Brewers, sports.sports_team.previously_known_as, Seattle Pilots)}
B. Miller Park {relevant facts: (Milwaukee Brewers, sports.professional_sports_team.owner_s, Mark Attanasio), (Milwaukee Brewers, sports.sports_team.arena_stadium, Miller Park)}
C. Milwaukee {relevant facts: (Milwaukee Brewers, sports.professional_sports_team.owner_s, Mark Attanasio), (Milwaukee Brewers, sports.sports_team.location, Milwaukee)} |
| DualR-w.o.-chain | Reference answers:
A. Seattle Pilots (correct probability: 0.7)
B. Miller Park (correct probability: 0.285)
C. Milwaukee (correct probability: 0.015) |

