# OpenReview forum: "Dual Reasoning: A GNN-LLM Collaborative Framework for Knowledge Graph Question Answering"
_CPAL.cc/2025/Proceedings_Track — CPAL 2025 (Proceedings Track) Poster_

### Official Review · Reviewer_kqFB · 2025-01-03

**Rating:** 6
**Confidence:** 3

**Review:**

This paper introduces Dual-Reasoning (DualR), a novel GNN-LLM collaborative framework for Knowledge Graph Question Answering inspired by dual-process theory in cognitive science.  It shows that by combining an LLM-empowered GNN for explicit knowledge extraction with a knowledge-enhanced prompting strategy for LLM-based answer determination, DualR achieves state-of-the-art performance on benchmark KGQA datasets. The innovative integration of GNNs and LLMs for reasoning over knowledge graphs presents a significant advancement in the field.

The reviewer finds the combination of GNN and LLM pretty interesting and the performance improvement seem to be natural, hence recommending acceptance at CPAL.

---

### Official Review · Reviewer_U5Kn · 2025-01-11
**This paper proposed a novel reasoning method for LLM.**

**Rating:** 6
**Confidence:** 1

**Review:**

This paper discusses a novel reasoning approach called dual reasoning, which is motivated by the dual-process theory. The paper's statements are clear, and the experimental results support the authors' claims. However, to be honest, the reviewer is not familiar with this topic and is therefore unable to provide detailed feedback on this paper. As a result, I believe the AC may disregard my comments.

---

### Official Review · Reviewer_Sn7p · 2025-01-12

**Rating:** 8
**Confidence:** 4

**Review:**

**Summary**
This paper introduces DualR, a novel framework that combines GNNs and LLMs for knowledge graph question answering. DualR employs an LLM-empowered GNN module to extract high-quality reasoning chains from knowledge graphs, which are then used to guide LLM reasoning through a knowledge-enhanced multiple-choice prompt format.

**Strengths**
* The technical approach is well-designed, with the GNN component providing explicit, structured reasoning capabilities while leveraging LLM knowledge for semantic understanding.
* The ablation studies thoroughly validate the importance of different components

**Weaknesses**
* The paper does not fully explore how closely the GNN and LLM components map to System 1 and System 2 characteristics beyond surface-level analogies.
* The approach assumes answers must be entities within the knowledge graph, which may limit applicability to more QA scenarios

---

### Meta-Review · Area_Chair_c6gN · 2025-02-05

**Recommendation:** Accept (Poster)
**Confidence:** 4

**Metareview:**

This paper discusses a reasoning approach motivated by the dual-process theory from cognitive science. It shows that it could bring substantial performance improvements by combining an LLM-empowered GNN for explicit knowledge extraction with a knowledge-enhanced prompting strategy for LLM-based answer determination. Reviewers acknowledge the innovation and significance of this research.

---

### Decision · Program_Chairs · 2025-02-11

Accept (Poster)